# Activity of Patchouli and Tea Tree Essential Oils against Staphylococci Isolated from Pyoderma in Dogs and Their Synergistic Potential with Gentamicin and Enrofloxacin

**DOI:** 10.3390/ani13081279

**Published:** 2023-04-07

**Authors:** Małgorzata Anna Szewczuk, Sławomir Zych, Nicola Oster, Jolanta Karakulska

**Affiliations:** 1Department of Monogastric Animal Sciences, Faculty of Biotechnology and Animal Husbandry, West Pomeranian University of Technology in Szczecin, 29 Klemensa Janickiego, 71-270 Szczecin, Poland; 2Laboratory of Chromatography and Mass Spectroscopy, Faculty of Biotechnology and Animal Husbandry, West Pomeranian University of Technology in Szczecin, Klemensa Janickiego 29, 71-270 Szczecin, Poland; 3Department of Microbiology and Biotechnology, Faculty of Biotechnology and Animal Husbandry, West Pomeranian University of Technology in Szczecin, Piastów 45, 70-311 Szczecin, Poland

**Keywords:** essential oils, patchouli, tea tree, staphylococci, pyoderma, MIC, checkerboards

## Abstract

**Simple Summary:**

The discovery of antibiotics was a breakthrough in medicine. However, bacterial defense mechanisms driven by genetic variation resulted in resistance to these compounds relatively quickly. Moreover, new classes of antibiotics have not been developed for 30 years. Within the European Union, the EU Parliament and Council Regulation No. 2019/6, which concerns veterinary medicinal products, is currently in force. The current goal is to reduce the use of antibiotics and to stop the rise of drug resistance in bacteria because such antimicrobial resistant organisms can be transmitted to humans through the consumption of animal products or direct contact with animals (dogs, cats, etc.). For this reason, there is a growing interest in essential oils (EOs). As natural mixtures (usually of terpenes and their derivatives), they may consist of about 20–60 components with 1–3 dominant component(s). An important feature of EOs is their hydrophobicity, which allows them to react with lipids present in bacterial cell membranes and mitochondria, disrupting the functioning of cell structures and consequently making them more permeable to other components or antibiotics. In the present manuscript, the activity of two EOs (patchouli and tea tree) was assessed, and their interaction with gentamicin and enrofloxacin was studied.

**Abstract:**

In this paper, we show the effect of some essential oils (EOs) on staphylococci, including multidrug-resistant strains isolated from pyoderma in dogs. A total of 13 *Staphylococcus pseudintermedius* and 8 *Staphylococcus aureus* strains were studied. To assess the sensitivity of each strain to the antimicrobial agents, two commercial EOs from patchouli (*Pogostemon cablin*; PcEO) and tea tree (*Melaleuca alternifolia*; MaEO) as well as two antibiotics (gentamicin and enrofloxacin) were used. The minimum inhibitory concentration (MIC) followed by checkerboards in the combination of EO-antibiotic were performed. Finally, fractional inhibitory concentrations were calculated to determine possible interactions between these antimicrobial agents. PcEO MIC ranged from 0.125 to 0.5 % *v*/*v* (1.2–4.8 mg/mL), whereas MaEO MIC was tenfold higher (0.625–5% *v*/*v* or 5.6–44.8 mg/mL). Gentamicin appeared to be highly prone to interacting with EOs. Dual synergy (38.1% of cases) and PcEO additive/MaEO synergism (53.4%) were predominantly observed. On the contrary, usually, no interactions between enrofloxacin and EOs were observed (57.1%). Both commercial EOs were characterized by natural composition without artificial adulteration. Patchouli and tea tree oils can be good alternatives for treating severe cases of pyoderma in dogs, especially when dealing with multidrug-resistant strains.

## 1. Introduction

Antibacterial therapies are mainly based on antibiotics. However, they are not always effective and can sometimes be invasive or cause side effects (hair or hearing loss, diarrhea, irritability, lack of appetite, etc.). Moreover, they increasingly encounter antibiotic resistance, which is a therapeutic and economic problem.

Purulent dermatitis (pyoderma) is the most common bacterial skin disease of dogs accompanying other dermatological problems, manifesting as a complication of the underlying disease, such as allergies (food allergy, atopic dermatitis, allergy to flea bites), internal diseases (hypothyroidism, adrenal hyperfunction), seborrhea and inflammation of the sebaceous glands, parasites (*Demodex canis*, scabies, etc.), hormonal fluctuations, anatomical predispositions (e.g., skin folds), or abnormal functioning of the immune system [1]. Puppies that have not yet developed a level of immunity and older dogs or steroid-treated individuals are the most vulnerable [2]. Pyoderma is much more common in dogs with short coats than in those with longer hairs, where the dense hair and undercoat provide a better barrier against bacterial penetration [3].

The symptoms that occur in pyoderma are varied and depend on the type of inflammation, the area of the skin and the intensity of the disease. The most common include erythema, blisters, itching, hair loss (alopecia), ulceration, coat and skin discoloration, scabs, pustules, and purulent lesions. It is also possible that skin lesions have an endocrine basis with other symptoms, such as lethargy, weight gain, or excessive thirst [4].

In the case of pyoderma in dogs, the most commonly isolated pathogen is a Gram-positive coccus classified as a *Staphylococcus pseudintermedius*. *S. pseudintermedius* is believed to colonize the skin and mucous membranes in small numbers in 80% of healthy dogs. Bacteria less commonly found in purulent lesions include other coagulase-positive staphylococci (*S. aureus* or *S. schleiferi* subsp. *coagulans*), Gram-negative bacilli—such as *Pseudomonas aeruginosa*, *Proteus* spp., and *Escherichia coli*—or yeast-like fungi (*Malassezia* sp., *Candida* sp.) [5]. These microorganisms are the natural commensal microflora of the skin in dogs; nevertheless, when they are abundant and the animal’s immune system declines, they can be the cause of skin lesions.

Antibiotics (and other antimicrobial agents not classified as antibiotics) used in the treatment of pyoderma in dogs should be characterized by a broad spectrum of action and high efficacy against the abovementioned microorganisms, i.e., mainly *S. pseudintermedius*. The drug must reach high concentrations in the skin and have as few side effects as possible. The most important attribute is strong bactericidal activity. Cephalosporins (e.g., first-generation cephalexin or third-generation cefovecin (Convenia)), fluoroquinolones (enrofloxacin, marbofloxacin, ciprofloxacin), aminoglycosides (amikacin, gentamicin), lincosamides (clindamycin), and amoxicillin/clavulanic acid are among the most commonly used antibiotics in the control of purulent dermatitis in dogs [6].

In recent years, there has been growing antibiotics resistance in staphylococci isolated from dogs. In addition to *S. aureus* (methicillin-resistant *Staphylococcus aureus*—MRSA), methicillin-resistant *Staphylococcus pseudintermedius* (MRSP) strains have appeared. This means resistance of these bacteria to antibiotics included in the β-lactam group. Additionally, these pathogens tend to be multidrug-resistant, which poses a problem in selecting the correct antibiotic during treatment [7]. They occur in the pharynx, nasal cavity, rectum, and periosteal area as asymptomatic carriage. Moreover, these bacteria are often isolated from dog bite wounds [8].

Because of this, there has been growing interest in essential oils (EOs) and their use in medicine, cosmetology, and the food industry. EOs are obtained from various plant materials (leaves, buds, fruits, flowers, herbs, branches, bark, wood, roots, and seeds) via steam distillation through their maceration with fats or pressing [9]. Essential oils are volatile, liquid, transparent or rarely colored, and soluble in fat and organic solvents. As natural mixtures of an extraordinarily complex nature, they can consist of up to 100–200 chemical compounds in a wide variety of concentrations: several are present in high concentrations (a total of 20–70%) compared to other components (trace amounts). The amount varies depending on the part and species of the plant. They are chemical derivatives of terpenes and terpenoids [10].

Despite a number of studies on the composition of individual oils, detailed knowledge of their mechanism of action is still limited. Of particular importance is determining the effects of EOs on various microorganisms, especially how they act in combination with other antimicrobial compounds [11].

Essential oils are believed to have important antiseptic, antibacterial, antiviral, antioxidant, antiparasitic, antifungal, and insecticidal activities [12]. An important characteristic of EOs is hydrophobicity, which allows them to dissociate from the lipids present in the bacterial cell membrane and mitochondria, making them more permeable by disrupting cell structures. This ultimately results in bacterial cell death due to the leakage of critical molecules and ions from the bacterial cell at a high rate [13]. EOs can thus serve as a powerful tool for inhibiting the growing phenomenon of bacterial resistance [14]. The overall concept of some antimicrobial synergy is based on the principle that combination of two or more antimicrobial agents may enhance efficacy, reduce/decrease toxicity or side effects of one of agent used, increase bioavailability, lower the dose of, e.g., antibiotics, and reduce the advance of antimicrobial resistance [15]. New and highly effective antimicrobial combinations of drugs that contain natural product(s) have recently become a research priority.

The aim of the study was to evaluate the antimicrobial activity of patchouli and tea tree essential oils applied alone and in combination with gentamicin and enrofloxacin as an alternative in the treatment of purulent skin inflammation in dogs against *S. pseudintermedius* and *S. aureus*, especially in regard to multidrug-resistant isolates.

## 2. Materials and Methods

### 2.1. Bacterial Strain Origin and Identification

All strains were isolated and collected in the veterinary laboratory (West Pomerania, Szczecin, Poland) during routine tests of swabs/skin scrapings from acute pyoderma cases in the years 2019–2021. Strains were then systematically banked (VIABANK™, MWE Medical Wire, Corsham, UK) in order to produce autovaccines and kept in a frozen state (≤−30 °C) until research (no animals were directly involved in this experiment). A total of 12 *S. pseudintermedius* and 7 *S. aureus* strains were archived. Additionally, an *S. aureus* reference strain (ATCC 25923, KWIK-STIK™ Microbiologics, Argenta, Poznan, Poland) and an *S. pseudintermedius* ED99 type strain (lab collection) were used as an internal control of the entire study. For the purposes of the research presented in the manuscript, the strains were revived onto blood agar, mannitol salt agar (Oxoid, Argenta, Poznan, Poland), and STAPH chromagar (GRASO, Starogard Gdanski, Poland) and incubated overnight at 37 °C.

After bacterial growth, isolates were identified on the basis of their ability to ferment mannitol (GRASO, Starogard Gdanski, Poland), Polymyxin B resistance, and ability to form fibrin clots in rabbit plasma (Biomed, Cracow, Poland). The Staphaurex™ Plus Latex Agglutination Test (Remel, ThermoFisher Scientific, Waltham, MA, USA) was also performed. After initial selection, a multiplex polymerase chain reaction (M-PCR) method described by Sasaki et al. [16] was performed to differentiate of coagulase-positive staphylococci (CoPS). By using this method, seven species of CoPS were preliminary differentiated based on the size of the PCR product after amplification of the conserved regions of the thermonuclease (*nuc*) gene. In addition, the presence of the *mecA* and *blaZ* genes were also tested according to Ruzauskas et al. [17]. In this case, the major genetic determinants of resistance to ß-lactam antibiotics were tested.

### 2.2. Antimicrobial Susceptibility Testing

The disk diffusion method was used to determine which antimicrobial agent will inhibit the growth of the selected staphylococci according to the CLSI M100 31st ed. [18] and VET01S 5th ed. [19] recommendations. The following commercial disks (OXOID, Argenta, Poznan, Poland) were used: penicillin G (5 μg), amoxicillin (10 μg), amoxicillin with clavulanic acid (20 + 10 μg), cefalexin (30 μg), doxycycline (30 μg), oxytetracycline (30 μg), trimethoprim/sulfamethoxazole (1:19; 25 μg), neomycin (30 μg), gentamicin (10 μg), amikacin (30 μg), enrofloxacin (5 μg), marbofloxacin (5 μg), ciprofloxacin (10 μg), and polymyxin B (300 U). In order to estimate potential methicillin resistance, an oxacillin disk (1 μg; resistance with the zone of inhibition ≤17 mm recommended for *S. pseudintermedius*) and a cefoxitin disk (30 μg; a surrogate for oxacillin recommended for *S. aureus* with the zone of inhibition ≤21 mm in regard to resistance) were used.

### 2.3. Antibiotics and Essential Oils Analysis

A freeze-dried gentamicin (OXOID, ThermoFisher Scientific, Waltham, MA, USA) was dissolved in deionized water to a final concentration of 256 mg/mL, becoming the basis for the appropriate two-fold dilutions in Mueller–Hinton broth (MHB) (GRASO, Gdansk, Poland) to achieve a final concentration ranging from 2.56 mg/mL to 0.01 μg/mL.

Enrofloxacin, as a ready-to-use solution for injection (Baytril™ One, 100 mg/mL suspended in 30 mg n-Butanol; Bayer Animal Health, UK), was purchased from Medivet (Szczecin, Poland). Similarly to above, the solution was initially diluted to concentration of 2.56 mg/mL, and then, two-fold dilutions were prepared.

Commercial essential oils (Organique/Avicenna, Wroclaw, Poland) from patchouli (*Pogostemon cablin*; PcEO) and tea tree (*Melaleuca alternifolia*; MaEO) were used in the study. Only to control the content of patchouli alcohol, a sample of Tisserand Aromatherapy brand of patchouli oil (First Natural Brands Ltd., Sayers Common, West Sussex, United Kingdom) was used in the HPLC-MS study as an internal control. Vials were stored at 4 °C in dark glass bottles. Dimethyl sulfoxide (DMSO) (Avantor, Gliwice, Poland) was used as an organic solvent for essential oils (twofold dilutions expressed as % *v*/*v* and mg/mL). A stock solution of the tested oils was prepared in a final concentration ranging from 10% to 0.001% *v*/*v*. The concentration was expressed in mg/mL depending on the individual density of the EO batch. In order to exclude an inhibitory effect of DMSO on growth of staphylococci, a concentration gradient of DMSO alone ranging from 0% to 50% (increase by 5%) was performed, and the survival of each staphylococci strain in this gradient was evaluated.

### 2.4. Activity of Antibiotics and Essential Oils against Staphylococci

#### 2.4.1. Individual MIC

To assess the sensitivity of each staphylococci strain to the antimicrobial agents under study, the minimum inhibitory concentration (MIC) of gentamicin, enrofloxacin, and both essential oils (PcEO and MaEO) against all *S. pseudintermedius* and *S. aureus* strains was individually determined via the serial dilution method using sterile 96-well plates (Wuxi Nest Biotechnology, Wuxi, China). Briefly, a decreasing concentration of antibiotics was successively added in the amount of 10 µL to each well containing 85 µL MHB in the rows of the 96-well microplate (causing an additional dilution of 1:10). In the case of EOs, 10 µL of decreasing-concentration, previously prepared stock solution of EOs was used in a similar arrangement as above (1:10; max DMSO content ≤ 10%). Next, the bacterial suspension (5 µL) at a concentration of 2.0 × 10^7^ CFU/mL (DEN-1 densitometer, BioSan, Józefów, Poland) was added to each well (final concentration approx. 1.0 × 10^6^ CFU/mL per well). The MIC was estimated after 24 h of incubation at 37 ± 1 °C. To avoid a false reading (especially at EO with artificial turbidity at the highest concentrations), a 10 µL of 0.01% resazurin (POL-AURA, Olsztyn, Poland) was added to each well. The color changed from blue to pink after an additional 3 h of incubation with resazurin at 37 ± 1 °C, indicating the presence of live bacteria in the well (which means that the antimicrobial agent was ineffective at the concentration tested). MIC was determined on the basis of the dark blue color appearance in the first well after any pink wells (corresponding to the smallest concentration of an antimicrobial agent capable of eliminating staphylococci). All experiments were performed in triplicate.

#### 2.4.2. Checkerboards

Knowing the individual effective concentrations of gentamicin, enrofloxacin and EOs, we extended our investigation to study the potential synergistic or antagonistic effect between those antimicrobial agents in the following four combinations: PcEO × gentamicin, PcEO × enrofloxacin, MaEO × gentamicin, and MaEO × enrofloxacin in the 96-well checkerboard.

Briefly, a mix of seven serial twofold dilutions of EO in rows (10 µL/well; horizontal orientation) and ten serial twofold dilutions of antibiotics in columns (10 µL/well; vertical orientation) was added to 75 µL/well of MHB for different oil × antibiotic combinations/well. The last row and penultimate column always contained only a single antimicrobial agent supplemented with the pure lacking opposite diluent (DMSO or ddH_2_O). The last column was reserved for positive and negative controls (wells contain only MHB and both diluents). Then, 5 µL of the particular bacterial suspension at a final concentration of 2.0 × 10^7^ CFU/mL was added to each well (final concentration approx. 1.0 × 10^6^ CFU/mL per well with the exclusion of negative controls which control the purity of MHB and diluents). Positive control confirms the vitality of the strain under conditions of maximum DMSO concentration in MHB. Incubation and readings were similar to the individual MIC. If there was an interaction, the best well was selected. Each checkerboard was performed in triplication.

#### 2.4.3. Fractional Inhibitory Concentrations

To determine possible interactions between antimicrobial agents, fractional inhibitory concentrations (FICs) were calculated according to van Vuuren and Viljoen [15] as follows:FIC(OxA) = MIC(OxA)/MIC(O)
FIC(AxO) = MIC(AxO)/MIC(A)
where:OxA—oil in combination with antibiotics    O—oil alone
AxO—antibiotics in combination with oil    A—antibiotics alone

The ΣFIC was then calculated for each test sample independently as the sum of the FIC:ΣFIC = FIC(OxA) + FIC(AxO)

The interpretation of possible interactions in vitro between antimicrobial agents was described as synergistic (ΣFIC ≤ 0.5), additive (0.5 < ΣFIC ≤ 1.0), noninteractive (1.0 < ΣFIC ≤ 4.0), or antagonistic (ΣFIC > 4.0).

### 2.5. Qualitative Analysis of the Composition of Essential Oils

Composition of both commercial essential oils were analyzed via high-performance liquid chromatography–mass spectrometry (HPLC-MS) technique. A reversed-phase Zorbax 2.1 × 50 mm Eclipse Plus C18 column (Agilent, Santa Clara, CA, USA) equipped with a guard column was used for the chromatographic separation. An Ultivo G6465B mass spectrometer (Agilent, USA) coupled to a chromatograph (1260 Infinity II Series Liquid Chromatograph, Agilent, USA) was used to detect and identify the constitutes according to mass-to-charge ratio (*m*/*z*) working in scanning mode (SCAN) followed by multiple reaction monitoring (MRM) mode.

The patchouli oil and tea tree oil were diluted using HPLC hypergrade acetonitrile—ACN (Supelco, Sigma Aldrich, Burlington MA, USA)—to prepare a concentration of 100 mg/mL. Then, the concentration of the oil was further rediluted to obtain lower dilutions for injection into the mass spectrometer (injection volume 1 μL). Mobile phase A was ddH_2_O containing 0.1% HCOOH (Formic Acid 98–100%, Suprapur, Merck, Germany), whereas mobile phase B was 100% ACN, also containing 0.1% HCOOH.

The source of electrospray ionization (ESI) operated in positive (M+H+ and other) and negative (M-H+ and other) modes. The triple quadrupole (QQQ) instrument operated under the following conditions: column temperature 25 °C, flow rate 0.3 mL/min, scan time (0.100 s to 0.500 s), fragmentator 5–120 V, collision energy interval (5.00–50.00 eV), and scanning range (100–750 *m*/*z*).

Whole system control and data acquisition were performed using MassHunter Acquisition Software ver. C.01.00 (Agilent, Santa Clara, CA, USA). The data obtained were analyzed using Qualitative Analysis Software ver. B.08.00 (Agilent, Santa Clara, CA, USA).

## 3. Results

The results of the drug resistance, tests, and PCR analysis of 21 strains under study are summarized in Table 1.

All coagulase-positive staphylococcal strains met the required criteria. *Staphylococcus aureus* ATCC 25923 and Sa 1–Sa 7 isolates have characteristics common to *S. aureus*, e.g., an immediate reaction in a high-specific latex test capability of fermenting mannitol. A typical PCR band of 359 bp fragment of the *nuc* gene was also obtained. On the contrary, other strains (Sps 1–Sps 12 and ED99) have always yielded a PCR band of 926 bp (according to Sasaki et al. [16], this fragment of the *nuc* gene is specific only to *S. pseudintermedius*) and negative results for the abovementioned tests. They also had quite a characteristic double hemolysis. Additionally, all the isolates were further identified via matrix-assisted laser desorption ionization time-of-flight mass spectrometry (MALDI-TOF-MS) (AniCon Labor GmbH, Germany) due to the collection of strains for autovaccines. Third-party test reports confirmed the species affiliation of all staphylococci under study.

Screening for drug resistance also revealed some patterns. Both reference strains, as well as Sa 4–Sa 7 and Sps 11, were susceptible to all antibiotics tested (except Polymyxin B, because *S. aureus* is naturally resistant to this antibiotic) and were *mecA-* and *blaZ* negative. Sps 9, Sps 10, and Sps 12 strains showed resistance only to single antibiotics (also *mecA*-negative; Sps 9 was *blaZ*-positive with resistance only to penicillins). In contrast, the remaining isolates (Sa1–Sa 3 and Sps 1–Sps 8) were resistant to the majority of antimicrobial agents examined, including cefoxitin/oxacillin. Moreover, only these isolates were simultaneously positive for the presence of the *mecA* gene (527 bp) and the *blaZ* gene (772 bp). Compiling this information, it can be concluded that isolates Sa1–Sa3 can be considered to represent a methicillin-resistant *S. aureus* (MRSA), whereas Sps 1–Sps 8 are considered methicillin-resistant *S. pseudintermedius* (MRSP). However, further in-depth studies are needed to confirm this hypothesis.

After preliminary analyses, the first dilution of essential oils was determined at levels of 1% (patchouli; density 0.966 g/mL) and 10% (tea tree; density 0.895 g/mL). A detailed susceptibility analysis to selected antibiotics (gentamicin, enrofloxacin) and essential oils (patchouli, tea tree) using the MIC method is summarized in Table 2.

In general, the MIC results for antibiotics were in agreement with the disk diffusion method. A group of staphylococci is especially notable for their high MIC values (high resistance) for both antibiotics (Sps 1 to Sps 8). The Sa3 staphylococcus strain was also characterized by significant dual resistance. One-way moderate or high resistance was also noted in the case of Sps 9 and Sps 12 (gentamicin) as well as Sa 1, Sa 2 and Sps 10 (enrofloxacin). The MIC values are also in agreement with the standards [18,19]. In the case of EOs, definitely more balanced results were obtained. For PcEO, MIC values ranging from 0.25 to 0.5% *v*/*v* (2.4–4.8 mg/mL) were recorded most often (81%; all *S. pseudintermedius* and antibiotic-resistant strains of *S. aureus*) followed by 0.125–0.25% *v*/*v*, which corresponds to a PcEO concentration of 1.2–2.4 mg/mL (19%; only for antibiotic-sensitive strains of *S. aureus*). The activity of MaEO was more varied regardless of drug resistance. Most often, the MIC ranged from 1.25–2.5% *v*/*v* (38% of cases; concentration 11.2–22.4 mg/mL) followed by 0.625–1.25% *v*/*v* (19%; 5.6–11.2 mg/mL), 1.25–5% *v*/*v* (19%; 11.2 ÷ 44.8 mg/mL) and 2.5–5 (14%; 22.4 ÷ 44.8 mg/mL). In summary, a tenfold stronger effect of PcEO than MaEO was noted. The DMSO content in the wells did not exceed 10%. However, our strains were still able to survive in 15–20% DMSO. This is consistent with the general knowledge of DMSO activity that the content of DMSO should not be more than 10–15%, while DMSO in amounts of 5–7.5% had no effect on MICs [20].

The essential oils were tested in combination with antimicrobial drugs against resistant (11 strains) and susceptible bacteria (10 strains), in order to check for their possible synergistic or antagonistic interactions using checkerboard method. Results can be seen in Table 3. For comparison, Appendix A presents the corresponding results when considering the concentration of EOs expressed in mg/mL instead of % *v*/*v*. Regardless of the unit chosen, identical interactions were obtained.

Gentamicin appears to be highly prone to interacting with EOs. In 53.4% of cases, the PcEO had additive effects on *S. aureus* and *S. pseudintermedius* while synergism was observed for MaEO. There was also a significant percentage of dual synergy results (38.1%). Even the worst synergy (∑FIC = 0.5) indicates that four-fold reduction in MICs of antibiotic and EO were observed. In the case of stronger synergies, a remarkable decrease in the MIC values of EOs was observed (e.g., Sps 3/tea tree eight-fold of tea tree and up to sixteen-fold of gentamicin). A double additive effect was noted for only two staphylococci (Sa 7 and Sps 10, however near synergy), probably by the fact that these two staphylococci were the most sensitive to gentamicin (MIC 0.0625–0.125 μg/mL), which may have limited the margin for possible interaction. There were no cases with neutral or negative interaction (even within triplicates).

Enrofloxacin in combination with patchouli or tea tree oil mostly acted independently and neutrally—no interactions were observed in 57.1% of cases. Other staphylococci reacted with little reproducibility. However, the PcEO had always an additive affect whereas MaEO acted quite randomly with enrofloxacin (synergy 9.5%, additive 14.3%, noninteractive 19.1%, respectively). Among them, two staphylococci (Sa 1 and Sps 1) are worthy of extra comment because of their origin: both are isolated from the most severe cases of canine pyoderma, and, surprisingly, the best combination of interaction was obtained for them: additive (PcEO) and synergy (MaEO).

### HPLC-MS Analysis

Commercial EOs are available in diluted, highly concentrated, and—rarely—in undiluted forms. Often the price of such specifics reveals its level of purity or adulteration. An attempt was therefore made to make a preliminary assessment of the composition of the oils used in the study. The volume equivalent to a concentration of 0.1% *v*/*v* of each essential oil in the MIC was examined.

According to available gas chromatography–mass spectroscopy (GS-MS) analysis of patchouli oil, the presence of up to 30 volatile substances was identified but the main components are: patchouli alcohol (average 20–45% but sometimes up to 72%; molar mass 222.4 g/mol), followed by pogostol (222.4 g/mol; 0.2–6%), pogostone (224.3 g/mol; specific to *P. cablin* 0.1–27%), norpatchoulenol (206.3 g/mol; 0.1–4%), patchoulene (~8%), seychellene (~6%), α- and δ-guaiene (~18% of each), caryophellene (~8%) and bulnesene (3–23%) (204.4 g/mol of each) [21,22,23]. The nonvolatile chemical profile of PcEO was revealed for the first time using HPLC-Q-TOF-MS by Xie et al. [24], by whom an additional 73 nonvolatile constituents (i.e., 33 flavonoids, 21 organic acids, 9 phenylpropanoids, 4 sesquiterpenes, 3 alkaloids, and 3 other types of compounds) were identified and characterized (pachypodol was most abundant at 344.3 g/mol; other compounds have a molar masses usually greater than 300). In this manuscript, the scan range of patchouli oil was set from *m*/*z* 200 to 230 (covering the aforementioned volatile components), and the result is presented in Figure 1.

To quantify patchoulol using GC-MS, *m*/*z* = 41, 55, 83, 98, 125, 138, 161, 179, 189, 207, and 222 were usually selected as the diagnostic ions (major ions are underlined; [23,25,26]. However, there are no significant data on how patchouli alcohol behaves in the mobile phase using liquid chromatography–mass spectroscopy (LC-MS) studies. Of some surprise in the presented study was the fact that there were no clear peaks, with masses ranging from 221–223 *m*/*z*, which should correspond to patchouli alcohol [222 ± H] (marked with an asterisk in Figure 1). For this purpose, the Avicenna EO was compared to a sample of some other essential oil from a highly acclaimed brand—Tisserand EO, an expert in sourcing and blending 100% natural pure essential oils since 1974. As can be seen in Figure 1, both chromatograms and mass spectra are almost identical. In both cases, the most abundant is the peak at *m*/*z* 219.2 (100% of both abundance, 1.35 × 10^6^ and 1.05 × 10^6^, respectively), which may be a sought-after oxygenated sesquiterpene: patchouli alcohol. Some of the PcEO sesquiterpene hydrocarbons (SQHCs), e.g., α-patchoulene and β-patchoulene are suspected to be artifacts formed through the dehydration of patchoulol and subsequent Wagner–Meerwein rearrangements during steam distillation [27]. When patchoulol is dehydrated ([M+H-H_2_O]^+^ resulted in mass *m*/*z* 205), depending on the conditions, various mixtures of patchoulenes and other rearranged hydrocarbons may be obtained (e.g., resulted in *m*/*z* 219.2). The neutral loss of -CO, H_2_O, -OCH_3_, or -CH_3_ was commonly observed in MS spectra. The nature of the patchoulol changes that are occurring during LC-MS remains to be explained. The second-most frequently recorded peak was *m*/*z* 205.2 (chromatogram abundance 1.05 × 10^6^ with Abund % at 81.1 and 0.95 × 10^6^ with Abund % at 91.25, respectively), which corresponds to an extensive and diverse group of sesquiterpenes of equal mass 204.36 g/mol ([M+H]^+^), including patchoulenes, guaienes, seychellenes, and bulnesenes, although a certain percentage here may be dehydrated patchoulol. Both essential oils noticeably vary in the third peak: *m*/*z* 225.2—it most likely refers to pogostone ([M+H]^+^). The Tisserand EO seems to be richer in this component compared to the Avicenna EO, but this observation can only be confirmed by a quantitative study. Pachypodol (*m*/*z* 343.3, [M-H]^+^) and several other components were also detected at a low level, which may indicate the natural origin of the essential oils.

In the case of tea tree essential oil, the chemical composition of MaEO may be extremely variable, e.g., depending on chemotype (this means that several groups exist within a population of one plant species with the same morphological features differing in compositions of their products), and over 220 chemicals have been identified [28]. Essential oil of *Melaleuca* terpinen-4-ol type is predominant, whereas in the composition, it should have terpinen-4-ol (35–48%; 154.25 g/mol), γ-terpinene (14–28%; 136.23 g/mol), α-terpinene (6–12%; 136.23 g/mol), 1,8-Cineole/eucalyptol (0.01–10%; 154.25 g/mol), α-pinene (1–4%; 136.23 g/mol), p-cymene (0.5–8%; 134.22 g/mol), terpinolene (1.5–5%; 136.23 g/mol), α-terpineol (2–5%; 154.25 g/mol), sabinene (0.01–3.5%; 136.23 g/mol) (according to ISO 4730:2017-02 [29]). Mass spectra (range from 133 to 160, extremely specific) for the tea tree essential oil used for research (MIC and checkerboards) are summarized in Figure 2.

A high abundance of compounds with a mass-to-charge of 137 *m*/*z* was observed, which corresponds to a group of several monoterpenes with a mass of 136.23 g/mol ([M+H]^+^; abundance greater than 1.9 × 10^5^). A two-fold lower abundance (approx. 1.0 × 10^5^) was observed for compounds with a mass of 153 *m*/*z*, which probably corresponds to the neutral loss of hydrogen in terpinen-4-ol, eucalyptol, or—less possibly—α-terpineol (154.25 g/mol; [M-H]^+^). In contrast, the third peak (135 *m*/*z*; abundance approx. 1.0 × 10^5^) can be either p-cymene ([M+H]^+^ from 134.22 g/mol) in lesser amounts or cases of hydrogen loss in a monoterpenes group. The presence of other compounds that are not natively present in MaEO (e.g., sesquiterpenes) and adulteration with fragrance compositions of synthetic origin (linalool, citronellol, etc.) were not found. In conclusion, the presence of the main MaEO-specific compounds (terpinen-4-ol and monoterpenes) was confirmed.

## 4. Discussion

Long-term antibiotic treatments may increase the risk of selecting for multidrug-resistant bacteria, one of the most relevant current threats to public health. Antimicrobial-resistant organisms can be transmitted to humans and other animals in the European Union and other countries through the consumption of products of animal origin, by direct contact with animals or humans, or by other means (Regulation EU No 2019/6) [30]. Alternative therapies, including essential oils (EOs), have become very popular as natural remedies in veterinary medicine. The objective of this study was the establishment of novel approaches to conventional therapies using selected EOs for the treatment of canine skin disorders. The efficacy of EOs in inhibiting a variety of classical and opportunistic pathogens depends on the plant part (e.g., leaf, flower, or bark), origin (e.g., country), seasonal variations, the method of extraction of the essential oil, the procedure used in the antimicrobial assays (e.g., different broth), and the target microbial isolate [31]. Different staphylococcal species (incl. *S. pseudintermedius*) isolated from canine dermatitis were examined in study by Ebani et al. [32]. Among them, oregano (*Origanum vulgare* L.) and thyme (*Thymus vulgaris* L.) EOs resulted highly active against all staphylococcal strains tested. The research conducted by Nocera et al. [7] aimed to test in vitro the antimicrobial activity of 11 EOs (e.g., cinnamon or eucalyptus) against four methicillin-resistant *Staphylococcus pseudintermedius* (MRSP) and four methicillin-susceptible *S. pseudintermedius* (MSSP) pyoderma-associated clinical isolates. The obtained findings demonstrated a clear in vitro efficacy of some tested EOs against both MRSP and MSSP strains isolated from dogs. Unfortunately, neither study included both PcEO and MaEO.

Patchouli essential oil (PcEO) is obtained by steam distillation or hydrodistillation of the dried leaves of *Pogostemon cablin* (Blanco) Benth. (*Lamiaceae*). It has a unique woody odor—utilized in high-end fragrances and cosmetics [27]. This plant originated in Southeast Asia, Madagascar, India, Brazil, Japan, and China, but 90% of patchouli oil around the world is supplied from Indonesia [21]. Given its multicomponent nature, PcEO is also a part of a traditional Chinese medicine that has been used for the treatment of many ailments for centuries, e.g., to treat colds, nausea, fever, headache, and diarrhea [24,33]. Biofilms formed by bacteria are associated with highly enhanced resistance against antimicrobial agents, resulting in therapy failure. However, the PcEO may significantly inhibited the initial adherence phase of *S. aureus* biofilm development [34].

Using the disk diffusion method, Karimi [35] revealed that freshly hydrodistilated Philippine patchouli oil was found to be active only against the Gram-positive bacteria (*Staphylococcus aureus* ATCC 25923 and other *Staphylococcus* sp., *Bacillus* sp., and *Streptococcus* species). Moreover, both hospital and community clinical human isolates of methicillin-sensitive (MSSA) and methicillin-resistant (MRSA) *S. aureus* were sensitive to an MIC range of 0.03–0.06% *v*/*v*. High antistaphylococcal potential of PcEO has been confirmed for the group of 31 strains isolated from cases of bovine mastitis (MIC ranging from 0.01% *v*/*v* to 0.312% *v*/*v*) and the reference strain *S. aureus* PCM 2051 (0.625% *v*/*v*) using commercial oil (Pollena Aroma, Poland) [36]. The results of MICs performed by Yang et al. [37] showed that patchouli oil and its main components (patchouli alcohol and pogostone) have good antibacterial activities against *Staphylococcus aureus* ATCC2925 (MIC at the level of 4.5 mg/mL, 2 mg/mL and 1 mg/mL, respectively). Other studies showed that pure PcEO at > 40 μL/mL concentration reduced the growth of *Staphylococcus aureus* ATCC 6538 reference strain [38]. The high efficacy of PcEO against staphylococci (even multidrug-resistant ones) is also confirmed by the results in the presented manuscript, where the MIC ranged from 0.125 to 0.5% *v*/*v* that correspond to average PcEO concentrations of 1.2–4.8 mg/mL. To our knowledge, we also present the first study of the in vitro activity of PcEO against *S. pseudintermedius*.

Tea tree essential oil (MaEO) is the volatile oil obtained by distillation from the leaves and terminal branchlets of *Melaleuca alternifolia* (Maiden et Betche) Cheel [29]. As mentioned previously, the chemical composition of MaEO may be extremely variable depending on multiple parameters, such as biomass used (from wild or cultivated trees; only leaves or leaves plus terminal branchlets); chemotype (according to ISO 4730:2017-02); and mode of production (steam distillation versus hydrodistillation) [28].

The activity of tea tree oil against *S. aureus* is definitely better documented in the scientific literature. May et al. [39] reported MaEO MICs and minimum bactericidal concentrations (MBCs) of 0.12–0.5% for *S. aureus* (including MRSA) as well as time-kill studies in which essential oil with increased concentration of terpinen-4-ol displayed enhanced antimicrobial activity (4 h instead of 6 h for standard tea tree oil). The mechanisms of action of MaEO and three of its components—1,8-cineole, terpinen-4-ol, and α-terpineol—against *Staphylococcus aureus* ATCC 9144 were also investigated by Carson et al. [40]. At inocula of 5.0 × 10^5^ and 5.0 × 10^7^ CFU/mL, the MICs and MBCs were both 0.25% and 0.5% *v*/*v*, respectively. At an inoculum of 5.0 × 10^9^ CFU/mL, the MIC was 0.5% *v*/*v* and the MBC was 1% *v*/*v*. An identical result was obtained by Nelson [41], while similar results (0.12–0.5% *v*/*v*) against various methicillin resistant strains of seven species of *Staphylococcus*, including *S. aureus* were reported by Harkenthal et al. [42]. A slightly higher value for MSSA and MRSA was reported by Oliva et al. [43]—0.5–2% *v*/*v*. In our study, at inocula of approx. 1.0 × 10^6^ CFU/mL per well, the MIC ranged from 0.625 to 1.25% or 5.6–11.2 mg/mL (susceptible strains of *S. aureus*) and 2.5 to 10% *v*/*v* (multidrug-resistant strains), which refers to the MaEO concentration range of 22.4–89.6 mg/mL. A similarly high MIC (5 ÷ 10%) was reported by De Martini et al. [44] for 17 coagulase-positive Staphylococci (CoPS) isolated from canine *otitis externa* cases. In contrast to all the above results, extremely low MIC values were also reported. Mann and Markham [45] and Kumari et al. [46] reported an MIC value of 0.02–0.04% *v*/*v* of tea tree oil against *S. aureus*.

In a study performed by Meroni et al. [47], a total of 23 *S. pseudintermedius* strains were collected from clinical samples (pyoderma) from different dogs. The majority of them (61%) were resistant to more than three pharmacological categories and were classified as multidrug-resistant. These authors reported slightly higher or similar MICs (7.6 ± 3.2% *v*/*v*) to those presented in this manuscript (0.625 ÷ 5% *v*/*v*). In the study of Valentine et al. [48], a total of 25 MRSP and 25 MSSP isolates from dogs with skin and soft tissue infections were included. Tea tree oil has been shown to inhibit the growth of both types of *S. pseudintermedius* strains, with MICs ranging from 0.12 to 0.96% *v*/*v* and from <0.03 to 0.96% *v*/*v*, respectively. In an experiment by Han et al. [49], the antimicrobial effects of a topical skin cream (Korean Dara cream^®^) consisting of four natural oils (emu oil, jojoba oil, avocado oil, and tea tree oil) were evaluated through measurements of MIC (0.23% *v*/*v*) against three *S. pseudintermedius* isolates obtained from the nostrils of healthy dogs.

The antibacterial activity of pure terpinen-4-ol on *S. aureus* reference strains (ATCC 25923, ATCC 13150, NCTC 6571 and NCTC 29213) and clinical isolates was assessed by determining the MIC (0.25% *v*/*v* in most cases) and MBC (mostly 0.5 % *v*/*v*) in few studies in the scientific literature [50,51,52]. In the presented manuscript, the MIC values were 5–10-fold higher when a commercial MaEO was used instead of terpinen-4-ol. Probably the low content of terpinen-4-ol in Avicenna essential oil caused this result. According to Avicenna Oil’s official certificate of laboratory analysis (batch no. 27947), the terpinen-4-ol content was at the level of 40.4% (2.5 times lower than the pure reagent). The strong antibiofilm activity of terpinen-4-ol against *S. aureus* was found in the study by Cordeiro et al. [52] in a concentration-dependent manner even at sub-MIC concentrations. Moreover, in silico molecular docking analysis showed a possible interaction between terpinen-4-ol and penicillin-binding protein 2a (PBP2a), which is one of the main molecules involved in staphylococci resistance to beta-lactam drugs. Apart from beta-lactamases being used to inactivate the antibiotic, MRSA and MRSP strain resistance is mediated through the acquisition of a gene cassette containing the *mecA* gene, which encodes the low-affinity altered transpeptidase PBP2a [53]. Thus, the effective binding of terpinen-4-ol to the PBP2a protein and the consequent inhibition of its activity can be an effective adjuvant tool in the treatment of resistant strains [52]. These observations are in agreement with our study, as the presence of this gene was confirmed in many of the multidrug-resistant strains studied in this manuscript (Sa1–Sa3 and Sps1–Sps8). In addition, resistance to penicillins in staphylococci is also mediated by β-lactamases encoded by the *blaZ* gene [17]. Unfortunately, only interactions with the main representatives of aminoglycosides and fluoroquinolones (without identifying their individual resistance genes) but not beta-lactam drugs have been studied, so this may be a goal for future studies.

Essential oils, due to the small scale on which they are obtained, sometimes have a high price, which encourages dishonest manufacturers and distributors to adulterate them. The main methods of adulterating EOs are to dilute them with vegetable fat, mix them with cheaper EOs, or to add synthetic components to mimic the olfactory properties or the composition of the chemotype [54,55]. While in the first case, the natural fragrance bouquet does not change (it is only less intense), when other compounds are introduced (e.g., various terpenes), the fragrance impression is significantly modified. Such modifications can also affect the antimicrobial activity of EOs [56]. In our study, both commercial Avicenna-brand EOs tested via HPLC-MS were characterized by an appropriate composition that does not vary from the literature data. There was also no adulteration with additional fragrance compounds, such as linalool, citronellol, or limonene, which could alter their properties. Furthermore, the Avicenna patchouli oil had a similar composition to an essential oil from an established brand that was more expensive.

## 5. Conclusions

Both commercial EOs were characterized by natural composition without artificial adulteration. Patchouli and tea tree oils can be good alternatives for treating severe cases of pyoderma in dogs, especially when dealing with multidrug-resistant strains. Gentamicin, in comparison to enrofloxacin, appears to be highly prone to interacting with EOs. It is noticeable that patchouli oil had several times stronger of an effect on staphylococci compared to tea tree oil. However, tea tree oil is characterized by a stronger synergistic effect, having great potential as long as the natural products contain predominantly terpinen-4-ol. In the future, it is advisable to conduct tests with several oils from different manufacturers (and, e.g., variable chemotypes) as well as their interaction with beta-lactams to confirm the observations obtained.

## Figures and Tables

**Figure 1 animals-13-01279-f001:**
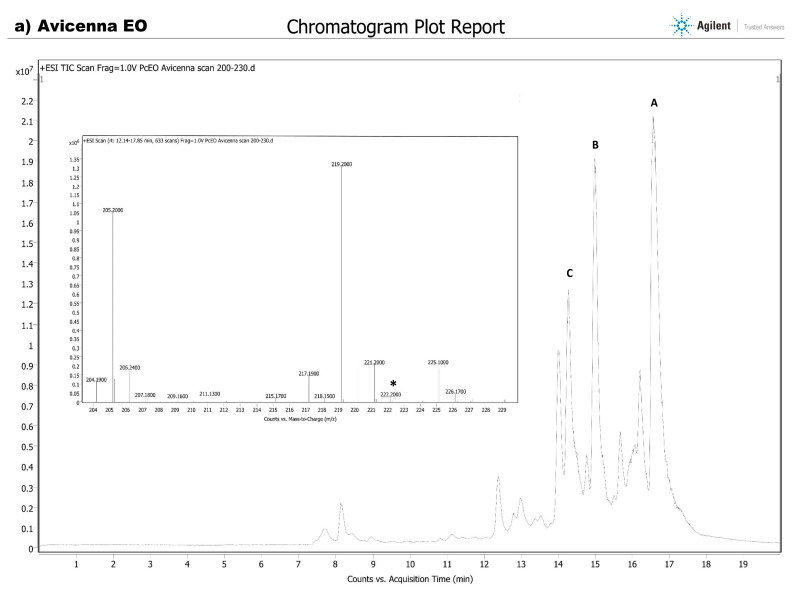
Chromatograms and mass spectra for the patchouli oils under study: Avicenna EO (**a**) and Tisserand EO (**b**). A—major peak for mass 205.2 *m*/*z*, B—major peak for mass 219.2 *m*/*z*, C—major peak for mass 225.2 *m*/*z*. *—potential position of nonmodified patchouli alcohol—patchoulol (222.4 *m*/*z*).

**Figure 2 animals-13-01279-f002:**
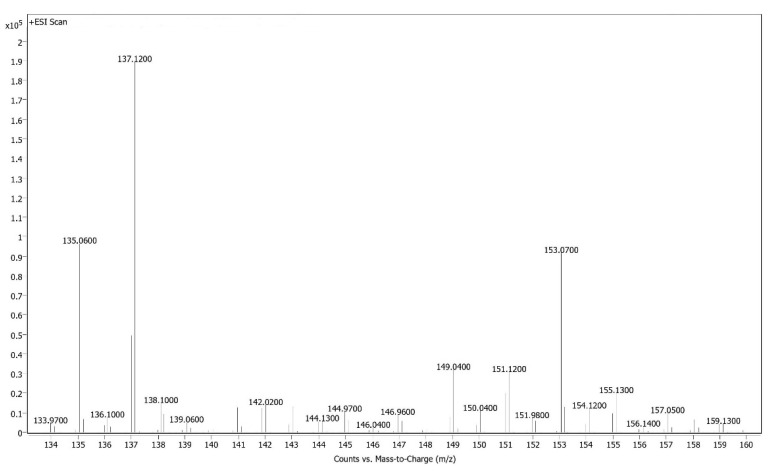
Mass spectra for the Avicenna tea tree essential oil under study.

**Table 1 animals-13-01279-t001:** Susceptibility testing (the disk diffusion method) and other differentiation tests for all staphylococci strains under study.

Antimicrobial Agents or Tests	Staphylococci																				
	Sa ATCC 25923	Sa 1	Sa 2	Sa 3	Sa 4	Sa 5	Sa 6	Sa 7	Sps ED99	Sps 1	Sps 2	Sps 3	Sps 4	Sps 5	Sps 6	Sps 7	Sps 8	Sps 9	Sps 10	Sps 11	Sps 12
Penicillins (penicillin G, amoxicillin)	S	R	R	R	S	S	S	S	S	R	R	R	R	R	R	R	R	R	S	S	S
amoxicillin potentiated with clavulanic acid	S	R	R	R	S	S	S	S	S	R	R	R	R	R	R	R	R	S	S	S	S
Cephalosporins (cefalexin)	S	R	R	R	S	S	S	S	S	R	R	R	R	R	R	R	R	S	S	S	S
tetracyclines (doxycycline, oxytetracycline)	S	R	I	S	S	S	S	S	S	R	R	R	I	I	R	S	I	S	S	S	S
sulfamethoxazole potentiated with trimethoprim	S	R	I	I	S	S	S	S	S	R	R	I	R	R	R	R	I	I	S	S	S
Aminoglycosides (neomycin, gentamicin, amikacin)	S	S	S	R	S	S	S	S	S	R	R	R	R	R	R	R	R	R	S	S	R
Quinolones (enrofloxacin, marbofloxacin, ciprofloxacin)	S	R	I	R	S	S	S	S	S	R	R	R	R	R	R	R	R	S	R	S	S
polymyxins (polymyxin B)	R	R	R	R	R	R	R	R	S	S	S	S	S	S	S	S	S	S	S	S	S
cefoxitin (disk diffusion method)	S	R	R	R	S	S	S	S	n/a	n/a	n/a	n/a	n/a	n/a	n/a	n/a	n/a	n/a	n/a	n/a	n/a
oxacillin (disk diffusion method)	[S]	[R]	[R]	[R]	[S]	[S]	[S]	[S]	S	R	R	R	R	R	R	R	R	S	S	S	S
M-PCR (bp)	359	359	359	359	359	359	359	359	926	926	926	926	926	926	926	926	926	926	926	926	926
*mec*A	-	+	+	+	-	-	-	-	-	+	+	+	+	+	+	+	+	-	-	-	-
*bla*Z	-	+	+	+	-	-	-	-	-	+	+	+	+	+	+	+	+	+	-	-	-
coagulase (rabbit serum)	+	+	+	+	+	+	+	+	+	+	+	+	+	+	+	+	+	+	+	+	+
Latex Agglutination Test	+	+	+	+	+	+	+	+	-	±	±	±	±	±	±	±	±	-	-	-	-
mannitol (anaerobic)	+	+	+	+	+	+	+	+	-	-	-	-	-	-	-	-	-	-	-	-	-

Sa—*Staphylococcus aureus*; Sps—*Staphylococcus pseudintermedius;* R—resistant; I—intermediate; S—susceptible; [S] or [R]—only in disk diffusion method, the results for oxacillin are based on the results for cefoxitin; bp—base pairs; n/a—not applicable; +—positive result; (-)—negative result; ±—trace granularity.

**Table 2 animals-13-01279-t002:** Minimum inhibitory concentrations (MICs) of gentamicin, enrofloxacin, and both essential oils (PcEO and MaEO) against all *S. pseudintermedius* and *S. aureus* strains.

Strain	MIC			
	Gentamicin	Enrofloxacin	Patchouli	Tea Tree
	μg/mL	μg/mL	% *v*/*v* (mg/mL)	% *v*/*v* (mg/mL)
Reference strain: *Staphylococcus aureus* ATCC 25923	0.5 ÷ 1	≤0.125	0.125 ÷ 0.25 (1.2 ÷ 2.4)	1.25 ÷ 2.5 (11.2 ÷ 22.4)
*Staphylococcus aureus* isolate Sa 1	0.5 ÷ 2	4 ÷ 8	0.25 ÷ 0.5 (2.4 ÷ 4.8)	2.5 ÷ 5 (22.4 ÷ 44.8)
*Staphylococcus aureus* isolate Sa 2	0.5 ÷ 1	2 ÷ 4	0.25 ÷ 0.5 (2.4 ÷ 4.8)	5 ÷ 10 (44.8 ÷ 89.6)
*Staphylococcus aureus* isolate Sa 3	8 ÷ 16	16 ÷ 32	0.25 ÷ 0.5 (2.4 ÷ 4.8)	2.5 ÷ 5 (22.4 ÷ 44.8)
*Staphylococcus aureus* isolate Sa 4	0.5 ÷ 1	0.125 ÷ 0.25	0.25 ÷ 0.5 (2.4 ÷ 4.8)	1.25 ÷ 2.5 (11.2 ÷ 22.4)
*Staphylococcus aureus* isolate Sa 5	0.25 ÷ 0.5	0.5 ÷ 1	0.125 ÷ 0.25 (1.2 ÷ 2.4)	1.25 ÷ 2.5 (11.2 ÷ 22.4)
*Staphylococcus aureus* isolate Sa 6	0.125 ÷ 0.25	0.125 ÷ 0.25	0.125 ÷ 0.25 (1.2 ÷ 2.4)	0.625 ÷ 1.25 (5.6 ÷ 11.2)
*Staphylococcus aureus* isolate Sa 7	0.0625 ÷ 0.125	0.032 ÷ 0.125	0.125 ÷ 0.25 (1.2 ÷ 2.4)	0.625 ÷ 1.25 (5.6 ÷ 11.2)
Type strain: *Staphylococcus pseudintermedius* ED99	0.5 ÷ 1	0.125 ÷ 0.25	0.25 ÷ 0.5 (2.4 ÷ 4.8)	1.25 ÷ 2.5 (11.2 ÷ 22.4)
*Staphylococcus pseudintermedius* isolate Sps 1	32 ÷ 64	32 ÷ 64	0.25 ÷ 0.5 (2.4 ÷ 4.8)	1.25 ÷ 5 (11.2 ÷ 44.8)
*Staphylococcus pseudintermedius* isolate Sps 2	32 ÷ 64	32 ÷ 64	0.25 ÷ 0.5 (2.4 ÷ 4.8)	1.25 ÷ 2.5 (11.2 ÷ 22.4)
*Staphylococcus pseudintermedius* isolate Sps 3	64 ÷ 128	32 ÷ 64	0.25 ÷ 0.5 (2.4 ÷ 4.8)	1.25 ÷ 2.5 (11.2 ÷ 22.4)
*Staphylococcus pseudintermedius* isolate Sps 4	32 ÷ 64	32 ÷ 64	0.25 ÷ 0.5 (2.4 ÷ 4.8)	0.625 ÷ 1.25 (5.6 ÷ 11.2)
*Staphylococcus pseudintermedius* isolate Sps 5	128 ÷ 256	32 ÷ 64	0.25 ÷ 0.5 (2.4 ÷ 4.8)	1.25 ÷ 5 (11.2 ÷ 44.8)
*Staphylococcus pseudintermedius* isolate Sps 6	64 ÷ 128	32 ÷ 64	0.25 ÷ 0.5 (2.4 ÷ 4.8)	2.5 ÷ 5 (22.4 ÷ 44.8)
*Staphylococcus pseudintermedius* isolate Sps 7	32 ÷ 128	64 ÷ 128	0.25 ÷ 0.5 (2.4 ÷ 4.8)	0.625 ÷ 1.25 (5.6 ÷ 11.2)
*Staphylococcus pseudintermedius* isolate Sps 8	64 ÷ 128	32 ÷ 64	0.25 ÷ 0.5 (2.4 ÷ 4.8)	1.25 ÷ 2.5 (11.2 ÷ 22.4)
*Staphylococcus pseudintermedius* isolate Sps 9	32 ÷ 64	≤0.125	0.25 ÷ 0.5 (2.4 ÷ 4.8)	1.25 ÷ 5 (11.2 ÷ 44.8)
*Staphylococcus pseudintermedius* isolate Sps 10	0.0625 ÷ 0.125	32 ÷ 64	0.25 ÷ 0.5 (2.4 ÷ 4.8)	1.25 ÷ 2.5 (11.2 ÷ 22.4)
*Staphylococcus pseudintermedius* isolate Sps 11	0.125 ÷ 0.5	≤0.125	0.25 ÷ 0.5 (2.4 ÷ 4.8)	0.625 ÷ 2.5 (5.6 ÷ 22.4)
*Staphylococcus pseudintermedius* isolate Sps 12	8 ÷ 32	0.25 ÷ 0.5	0.25 ÷ 0.5 (2.4 ÷ 4.8)	1.25 ÷ 5 (11.2 ÷ 44.8)

**Table 3 animals-13-01279-t003:** Checkerboard analysis with final interactions (best match within triplicate).

Antimicrobial Agent	Patchouli Oil				Tea Tree Oil			
	MIC_i_	MIC_c_	FIC	∑FIC [Interaction]	MIC_i_	MIC_c_	FIC	∑FIC (Interaction)
*Staphylococcus aureus* ATCC 25923								
oil (% *v*/*v*)	0.25	0.008	0.032	0.282 synergy	2.5	0.08	0.032	0.282 synergy
gentamicin (μg/mL)	1	0.25	0.25		1	0.25	0.25	
oil (% *v*/*v*)	0.125	0.125	1	2.0 none	1.25	1.25	1	2.0 none
enrofloxacin (μg/mL)	0.125	0.125	1		0.125	0.125	1	
*Staphylococcus aureus* Sa 1								
oil (% *v*/*v*)	0.5	0.125	0.25	0.75 additive	5	1.25	0.25	0.5 synergy
gentamicin (μg/mL)	2	1	0.5		0.5	0.125	0.25	
oil (% *v*/*v*)	0.5	0.25	0.5	1.0 additive	5	1.25	0.25	0.375 synergy
enrofloxacin (μg/mL)	8	4	0.5		8	1	0.125	
*Staphylococcus aureus* Sa 2								
oil (% *v*/*v*)	0.5	0.125	0.25	0.75 additive	10	1.25	0.125	0.25 synergy
gentamicin (μg/mL)	1	0.5	0.5		0.5	0.0625	0.125	
oil (% *v*/*v*)	0.25	0.25	1	2.0 none	5	5	1	2.0 none
enrofloxacin (μg/mL)	4	4	1		4	4	1	
*Staphylococcus aureus* Sa 3								
oil (% *v*/*v*)	0.5	0.125	0.25	0.5 synergy	2.5	0.625	0.25	0.5 synergy
gentamicin (μg/mL)	16	4	0.25		8	2	0.25	
oil (% *v*/*v*)	0.25	0.25	1	2.0 none	5	5	1	2.0 none
enrofloxacin (μg/mL)	32	32	1		32	32	1	
*Staphylococcus aureus* Sa 4								
oil (% *v*/*v*)	0.25	0.125	0.5	0.625 additive	2.5	0.32	0.128	0.253 synergy
gentamicin (μg/mL)	0.5	0.0625	0.125		0.5	0.0625	0.125	
oil (% *v*/*v*)	0.25	0.25	1	2.0 none	1.25	1.25	1	2.0 none
enrofloxacin (μg/mL)	0.25	0.25	1		0.125	0.125	1	
*Staphylococcus aureus* Sa 5								
oil (% *v*/*v*)	0.25	0.0625	0.25	0.75 additive	2.5	0.625	0.25	0.378 synergy
gentamicin (μg/mL)	0.25	0.125	0.5		0.25	0.032	0.128	
oil (% *v*/*v*)	0.25	0.25	1	2.0 none	1.25	1.25	1	2.0 none
enrofloxacin (μg/mL)	1	1	1		0.5	0.5	1	
*Staphylococcus aureus* Sa 6								
oil (% *v*/*v*)	0.125	0.016	0.128	0.384 synergy	1.25	0.16	0.128	0.256 synergy
gentamicin (μg/mL)	0.125	0.032	0.256		0.125	0.016	0.128	
oil (% *v*/*v*)	0.125	0.032	0.256	0.756 additive	1.25	0.625	0.5	1.0 additive
enrofloxacin (μg/mL)	0.125	0.0625	0.5		0.125	0.0625	0.5	
*Staphylococcus aureus* Sa 7								
oil (% *v*/*v*)	0.25	0.0625	0.25	0.506 additive	1.25	0.32	0.256	0.768 additive
gentamicin (μg/mL)	0.125	0.032	0.256		0.0625	0.032	0.512	
oil (% *v*/*v*)	0.125	0.0625	0.5	1.012 additive	0.625	0.625	1	2.0 none
enrofloxacin (μg/mL)	0.0625	0.032	0.512		0.125	0.125	1	
*Staphylococcus pseudintermedius* ED99								
oil (% *v*/*v*)	0.25	0.032	0.128	0.192 synergy	2.5	0.32	0.128	0.256 synergy
gentamicin (μg/mL)	0.5	0.032	0.064		0.5	0.064	0.128	
oil (% *v*/*v*)	0.125	0.008	0.064	0.564 additive	1.25	1.25	1	2.0 none
enrofloxacin (μg/mL)	0.25	0.125	0.5		0.125	0.125	1	
*Staphylococcus pseudintermedius* Sps 1								
oil (% *v*/*v*)	0.5	0.0625	0.125	0.375 synergy	5	0.625	0.125	0.375 synergy
gentamicin (μg/mL)	64	16	0.25		32	8	0.25	
oil (% *v*/*v*)	0.25	0.016	0.064	0.564 additive	1.25	0.08	0.064	0.314 synergy
enrofloxacin (μg/mL)	32	16	0.5		64	16	0.25	
*Staphylococcus pseudintermedius* Sps 2								
oil (% *v*/*v*)	0.5	0.125	0.25	0.75 additive	1.25	0.16	0.128	0.378 synergy
gentamicin (μg/mL)	64	32	0.5		32	8	0.25	
oil (% *v*/*v*)	0.25	0.016	0.064	0.564 additive	1.25	1.25	1	2.0 none
enrofloxacin (μg/mL)	64	32	0.5		64	64	1	
*Staphylococcus pseudintermedius* Sps 3								
oil (% *v*/*v*)	0.5	0.25	0.5	0.625 additive	1.25	0.16	0.128	0.191 synergy
gentamicin (μg/mL)	128	16	0.125		64	4	0.0625	
oil (% *v*/*v*)	0.5	0.032	0.064	0.564 additive	1.25	1.25	1	2.0 none
enrofloxacin (μg/mL)	64	32	0.5		64	64	1	
*Staphylococcus pseudintermedius* Sps 4								
oil (% *v*/*v*)	0.5	0.125	0.25	0.5 synergy	0.625	0.16	0.256	0.381 synergy
gentamicin (μg/mL)	64	16	0.25		32	4	0.125	
oil (% *v*/*v*)	0.5	0.5	1	2.0 none	1.25	1.25	1	2.0 none
enrofloxacin (μg/mL)	32	32	1		64	64	1	
*Staphylococcus pseudintermedius* Sps 5								
oil (% *v*/*v*)	0.5	0.125	0.25	0.5 synergy	5	1.25	0.25	0.281 synergy
gentamicin (μg/mL)	256	64	0.25		128	4	0.031	
oil (% *v*/*v*)	0.5	0.032	0.064	0.564 additive	1.25	0.625	0.5	1.0 additive
enrofloxacin (μg/mL)	64	32	0.5		64	32	0.5	
*Staphylococcus pseudintermedius* Sps 6								
oil (% *v*/*v*)	0.5	0.25	0.5	0.75 additive	2.5	0.32	0.128	0.159 synergy
gentamicin (μg/mL)	128	32	0.25		64	2	0.031	
oil (% *v*/*v*)	0.5	0.5	1	2.0 none	2.5	2.5	1	3.0 none
enrofloxacin (μg/mL)	32	32	1		64	128	2	
*Staphylococcus pseudintermedius* Sps 7								
oil (% *v*/*v*)	0.5	0.25	0.5	1 additive	0.625	0.16	0.256	0.381 synergy
gentamicin (μg/mL)	128	64	0.5		32	4	0.125	
oil (% *v*/*v*)	0.5	0.008	0.016	0.516 additive	1.25	0.625	0.5	1.0 additive
enrofloxacin (μg/mL)	64	32	0.5		128	64	0.5	
*Staphylococcus pseudintermedius* Sps 8								
oil (% *v*/*v*)	0.5	0.125	0.25	0.75 additive	1.25	0.16	0.128	0.191 synergy
gentamicin (μg/mL)	128	64	0.5		64	4	0.0625	
oil (% *v*/*v*)	0.5	0.5	1	2.0 none	1.25	1.25	1	2.0 none
enrofloxacin (μg/mL)	64	64	1		64	64	1	
*Staphylococcus pseudintermedius* Sps 9								
oil (% *v*/*v*)	0.5	0.125	0.25	0.375 synergy	5	1.25	0.25	0.375 synergy
gentamicin (μg/mL)	64	8	0.125		32	4	0.125	
oil (% *v*/*v*)	0.5	0.5	1	2.0 none	1.25	1.25	1	2.0 none
enrofloxacin (μg/mL)	0.125	0.125	1		0.125	0.125	1	
*Staphylococcus pseudintermedius* Sps 10								
oil (% *v*/*v*)	0.5	0.125	0.25	0.506 additive	1.25	0.32	0.256	0.512 additive
gentamicin (μg/mL)	0.125	0.032	0.256		0.0625	0.016	0.256	
oil (% *v*/*v*)	0.5	0.5	1	2.0 none	1.25	1.25	1	2.0 none
enrofloxacin (μg/mL)	64	64	1		32	32	1	
*Staphylococcus pseudintermedius* Sps 11								
oil (% *v*/*v*)	0.5	0.25	0.5	0.75 additive	2.5	0.625	0.25	0.378 synergy
gentamicin (μg/mL)	0.5	0.125	0.25		0.125	0.016	0.128	
oil (% *v*/*v*)	0.5	0.5	1	2.0 none	0.625	0.625	1	2.0 none
enrofloxacin (μg/mL)	0.125	0.125	1		0.125	0.125	1	
*Staphylococcus pseudintermedius* Sps 12								
oil (% *v*/*v*)	0.5	0.25	0.5	0.75 additive	5	1.25	0.25	0.5 synergy
gentamicin (μg/mL)	32	8	0.25		8	2	0.25	
oil (% *v*/*v*)	0.5	0.5	1	2.0 none	1.25	1.25	1	2.0 none
enrofloxacin (μg/mL)	0.5	0.5	1		0.5	0.5	1	

Sa—*Staphylococcus aureus;* Sps—*Staphylococcus pseudintermedius;* MIC_i_: MIC individually as MIC^(O)^ or MIC^(A)^; MIC_c_: MIC in combination as MIC^(OxA)^ or MIC^(AxO)^; FIC: fractional inhibitory concentrations; none = noninteractive.

## Data Availability

All data generated or analyzed during the study are included in this published article. The datasets used and/or analyzed in the current study are available from the corresponding author on reasonable request.

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
