# Peer review of "Activity of Patchouli and Tea Tree Essential Oils against Staphylococci Isolated from Pyoderma in Dogs and Their Synergistic Potential with Gentamicin and Enrofloxacin"

_animals, 2023, doi:10.3390/ani13081279_

Round 1

Reviewer 1 Report

In this manuscript, the use of patchouli and tea tree essential oils against methicillin-resistant staphylococci strains isolated from dogs was studied.

The following comments are made:

1. Lines 29, 32, 66-71, etc. The scientific names of the organisms are in italics, correct throughout the text

2. Lines 112 -115. Why use gentamicin and enrofloxacin to eliminate staphylococci, if these antibiotics are mainly used against Gram negative bacteria.

3. Line 134. Briefly describe PCR to differentiate between staphylococci.

4. Line 135. Briefly describe PCR for mecA and blaZ genes

5. Line 181. Put 7 as an exponent. How did you calculate the cfu?

6. Lines 252-257. It is part of the introduction, not the results.

7. Table 1. Correctly put n/a. Why didn't you test oxacillin and cefoxitin against staphylococci if their use against them is recommended? Determine resistance to oxacillin and cefoxitin

8. Table 1. What does M-PCR mean?

9. Line 268. A PCR band of what? What is the detected gene?

10. Line 270. The nucA gene is also characteristic of S. aureus.

11. Line 273. What were the results of the MALDI-TOF?

12. Lines 283-286. To know if they are MRSA or MRSP you have to do the oxacillin MIC

13. Table 2. You can put the best values in bold to visualize it better.

14. Line 318-326. How can you have different effects if it is the same bacteria and the same solutions? explain it

15. Line 338. For what use sell Commercial EOs?

16. Figure 1. What is the top and what is the bottom? Say it. What is the difference?

17. Line 416. Say which are the main isolated components

18. Line 453. Capitalizing Gram is a proper name.

19. In the conclusions you do not say anything about the interaction with antibiotics.

Reviewer 2 Report

This manuscript reports an interesting study about the activity of essential oils from patchouli and tea tree against Staphylococcus strains responsible for canine pyoderma.

The study is well organized, but some corrections are necessary. First of all, scientific names of bacteria, fungi, parasites and plants must be written in italic.

Lines 65 and 69: Gram-positive; Gram-negative

Line70: Malassezia (not Mallassezia)

line 142: were (not was)

line 181: 2.0 x 107 (7 in apice) CFU/ml

line 262: susceptible

Results: MIC of essential oils should be expressed also in μg/mL as for antibiotics. The comparison between μg/mL and %v/v is not possible.

Table3 should be reported as supplementary material.

Reviewer 3 Report

This study aimed to evaluate the antimicrobial activity of patchouli and tea tree essential oils, alone and in combination with antibiotics, against S. pseudintermedius and S. aureus, isolated from pyoderma in dogs. 

The manuscript is well written, the purpose of the study is very interesting and the research design is adequately described.

I have only a few observations to make:

  • the names of the bacteria should be written in italics.

  • line 136: when the authors mention the genes mecA and blaZ they should indicate the type of resistance they confer, to make it easier to understand.

  • Figures 1-2: the quality is a bit low and they are not very legible.

Round 2

Reviewer 1 Report

1. In the new version the changes made are not marked, it is necessary to detect the changes.

2. Lines 112 -115. So you should change the title of their manuscript since it specifically mentions MR staphylococci and pyoderma

3. Line 135. If it is a clarification of Material and Methods, why put it up for Discussion?

4. Line 181. Put in all the text CFU/mL

5. Line 252-257. All the information that is required to understand the experiments is always put in the introduction with their corresponding references.

6. Table 1. In the CLSI these are the recommended methods

7. Lines 283-286. So avoid saying MRSA  and MRSP strains and saying possibly throughout the text.

8. Line 318-326. How many bacterial strains did you test?

Reviewer 2 Report

Obviously, MIC can be expressed also as %v/v; however in this case,  μg/mL should be used as for antibiotics. Each production lot of commercial EOs has data to calculate the relative density.
